# Executive Functions in Children and Adolescents with Autism Spectrum Disorder in Family and School Environment

**DOI:** 10.3390/ijerph19137834

**Published:** 2022-06-26

**Authors:** Ana Gentil-Gutiérrez, Mirian Santamaría-Peláez, Luis A. Mínguez-Mínguez, Jessica Fernández-Solana, Jerónimo J. González-Bernal, Josefa González-Santos, Ana I. Obregón-Cuesta

**Affiliations:** 1Department of Health Sciences, University of Burgos, 09001 Burgos, Spain; agentil@ubu.es (A.G.-G.); jfsolana@ubu.es (J.F.-S.); jejavier@ubu.es (J.J.G.-B.); mjgonzalez@ubu.es (J.G.-S.); 2Department of Educational Sciences, University of Burgos, 09001 Burgos, Spain; 3Department of Mathematics and Computation, University of Burgos, 09001 Burgos, Spain; aiobregon@ubu.es

**Keywords:** autism, ASD, children, teenagers, fathers, executive functions, BRIEF-2

## Abstract

Autism spectrum disorder (ASD) is a neurodevelopmental disorder characterized by the presence of difficulties in communication and social interaction, often associated with deficits in executive functions (EF). The EF correct development is related to a more effective functioning in all its daily activities, while being associated with more efficient social relations. The objective of this research is to analyze the level of development of EF in children and adolescents with ASD in school and at home. This is a descriptive, cross-sectional, and multicenter study with 102 participants selected by non-probabilistic sampling, 32 parents of children with ASD, and 70 professionals in the field of education of students with ASD. The study confirms that although children and adolescents with ASD have problems in executive functioning, the perception of informants, parents, and education professionals is similar but not the same in the different contexts: school and home.

## 1. Introduction

Autism spectrum disorder (ASD) is a very heterogeneous complex neurodevelopmental disorder as for its etiologically, phenotypic diversity, and evolution variability in the different stages and presentation according to gender [1]. It is characterized by persistent difficulties in two dimensions, communication and social interaction, in addition to the manifestation of repetitive and restricted behaviors, interests, or activity patterns [2]. Severity degrees are related to the level of social and behavioral support required, including unusual behaviors in the field of sensory stimulation [2]. 

Executive function (EF) and information processing correspond to the degree of the skills and abilities aimed at functionally performing activities of daily living, and, consequently, to independence and autonomy in many natural environments.

There are several theories without consensus, but in general the term EF refers to a set of intentional and interrelated cognitive processes of higher order, aimed at issuing adaptive responses to deal with complex and novel situations that involve objectives for which we do not have an established and automated behavior [3,4,5]. Likewise, there are different dimensions, including cognitive inhibition, cognitive flexibility, working memory, self-control, and/or action monitoring [6].

EF correct development is associated with efficient social relations [5] and with a more effective functioning in all the activities carried out in daily life that provide meaning and purpose to life. 

On the one hand, a strong correspondence has been determined between EF deficits and social skills in people with ASD. These difficulties can contribute to additional behavioral and adaptive challenges during adolescence [7]. In contrast to social abilities, difficulties in executive processing may be likely to increase severely in childhood and adolescence [8].

On the other hand, the variability of challenges in the EF presented from early childhood by children and adolescents with ASD can contribute both to difficulties in communication and social interaction to restricted and repetitive behaviors, activities, and interests [9].

There is more information on the possible dysfunction at the EF level in people with autism in the domestic context than in the educational environment [10], denoting that the perception of teachers and parents suggests that they tend to underestimate the EF [6,11]. 

The evidence denotes that this group shows executive dysfunction, encountering difficulties in daily activities that involve planning, organization, or problem solving [5,12], as well as inhibitory control, working memory, or cognitive flexibility [4,12,13,14,15,16,17,18,19].

A correct evaluation of executive functioning is essential to propose effective interventions [3] and to provide the necessary support required by people with ASD to effectively face the demands of different social contexts [9]. Executive dysfunction is included in daily occupations, but further research is needed [20,21], so it is important to consider interindividual cognitive variability and differences in executive performance of people with ASD [22] to know the level of involvement in order to: firstly, shed more light on the development of executive functioning in childhood and adolescence in two very relevant social contexts in everyday life such as home [23] and school [24]; and secondly, to contribute to a better theoretical understanding of the heterogeneity of executive development, through the comparison of the perceptions of the people who know children with ASD best, providing a more complete and ecological profile [25] and, thirdly and finally, to understand what factors and to what extent they may be affecting executive processing, to favor the design and implementation of interventions according to evolutionary development [22].

The existence of an association between EF measured with the Behavioral Evaluation of Executive Function Questionnaire, second edition (BRIEF-2) in children and adolescents with symptoms of autism and adaptive functioning, internalization symptoms, and learning behaviors has been demonstrated [26]. It is important to study these behaviors from a developmental perspective and look at individual differences to implement good interventions [27].

The objective of this study is to analyze the development level of the EF in children and adolescents between 6 and 18 years old with autism in two well-differentiated social contexts, the educational and family, through the information related to EF obtained from parents of children with ASD and professionals in the field of education of students with ASD who acted as informants.

Therefore, the research question of this study is: Are there differences in the development of the EF in children and adolescents between 6 and 18 years old with autism in two well-differentiated social contexts: education and family?

## 2. Materials and Methods

### 2.1. Study Design

This is a cross-sectional, multicenter descriptive research study, with data obtained between September 2021 and January 2022 through an online questionnaire distributed to different ordinary educational centers and organizations that work with people with ASD in north-central Spain: Burgos, Cantabria, Madrid, Palencia, Salamanca, Valladolid and Vizcaya, which allowed the questionnaire to be delivered to the target population. 

### 2.2. Study Participants

After accepting the informed consent, 32 parents of children with ASD and 70 professionals in the field of education of students with ASD (participants in independent groups) collaborated in the study.

The inclusion criteria include on the one hand parents and legal guardians of children or adolescents aged 6 to 18 years diagnosed with autism in grade 1 and 2, according to the diagnostic criteria of the American Psychiatric Association [2]. The inclusion criteria also include professionals in the educational field working with children or adolescents aged 6 to 18 years, with the same ASD grade. These professionals, in addition, had to have contact with children at least once a week.

All the questionnaires with more than 10% of unanswered questions and questionnaires that, when correcting the validity scales, indicated an elevated risk of the data not being valid (level of caution or alert) were excluded from the study.

The Bioethics Committee of the University of Burgos approved the research, (Reference UBU 039/2020), respecting all the requirements established in the Declaration of Helsinki of 1975.

### 2.3. Procedure

This research was designed with the intention of answering the research question: Are there differences in the level of development of the EF in children and adolescents between 6 and 18 years old with autism in two well-differentiated social contexts, education and family?

The research presents a convenience sample that was available to researchers for the study to be developed.

After all the participants had signed the informed consent, the researchers obtained the data. BRIEF-2 tool was distributed to different educational ordinary centers, as well as to non-profit organizations that work with people with ASD; so they sent the questionnaire to the target population.

The BRIEF-2 Family questionnaire was completed by parents or relatives and the BRIEF-2 School by professionals in the educational field. So, the questionnaire collected information on the frequency of certain daily situations of children and adolescents in two social contexts: in the environment with parents (BRIEF-2 Family) and in school with education professionals (BRIEF-2 School). Its correction was made through the “TEA-corrige” platform (Trastorno del Espectro Autista-corrige, in Spanish).

After that, the questionnaires that fit the exclusion criteria were excluded from study.

Once the sample was obtained, statistical analysis was performed.

### 2.4. Instrument

The instrument used for the research was the Questionnaire of Behavioral Evaluation of Executive Function, second edition (BRIEF-2). It is completed through informants and it allows researchers to obtain an EF overview in everyday aspects from different informants’ perspectives and in natural contexts, such as home and school. It is a suitable tool for assessment in various clinical or social contexts [28].

Although the objective of (BRIEF-2) is not to diagnose, it may be useful within a more complete assessment by adding aspects of functionality in activities of daily living, as well as social communication of people with ASD [29]. 

It is a standardized and validated instrument that allows evaluating executive functions in children and adolescents aged between 5 and 18 years. The BRIEF-2 Family questionnaire can be completed by parents or relatives and the BRIEF-2 School by professionals in the educational field [28]. For this research, family informants used the BRIEF-2 Family version and professional informants used the BRIEF2 School version.

Both versions consist of 63 Likert response items and are performed in a time of 10 to 15 min. It also has 3 validity scales (infrequency, inconsistency, and negativity). The 63 items are distributed to make up the 9 clinical scales (inhibition, self-supervision, flexibility, emotional control, initiative, working memory, planning and organization, supervision of the task, and organization of materials). In addition, the tool has 3 indices that bring together the clinical scales (behavioral regulation index: inhibition and self-supervision, emotional regulation index: flexibility and emotional control and index of cognitive regulation: initiative, working memory, planning and organization, supervision of the task and organization of materials), as well as 1 index of executive function that results from the scores obtained in the other three indices. The correction process is conducted automatically through the online correction platform.

In the BRIEF-2, the assessment is made through typical scores that are expressed in T scores of the general population differentiated by sex, age range, and informants, with a mean of 50 and a standard deviation of 10 according to the reference sample. Where higher T scores correspond to a greater degree of executive dysfunction and with established cut-off points: ≥70 clinically significant elevation, 65–69 potentially clinical elevation, 60–64 mild elevation and 0–59 without apparent clinical significance.

This evaluation instrument has good psychometric properties, with a Cronbach’s alpha ranging from 0.73 to 0.90 in the BRIEF-2 Family [28].

### 2.5. Statistical Analysis

The sociodemographic particularities of the sample are detailed through a descriptive analysis in which the categorical variables are expressed by frequencies and percentages and the quantitative variables by means and standard deviations. Using the Saphiro-Wilk test, it was found that the sample did not conform to normal (*p* > 0.05), so non-parametric tests were performed. To check if there is a difference in the EF between the family environment and the school professional environment, the Mann–Whitney U statistical test was performed. Correlations were also made between each of the regulatory indices and the different clinical scales and with the children’s and adolescents’ age. A comparison by sex was also made.

Statistical analysis was performed with SPSS version 25 software (IBM-Inc, Chicago, IL, USA) and G*POWER software. For the analysis of statistical significance, a *p*-value < 0.05 was established. 

## 3. Results

A total sample of 102 subjects participated as informants about the EF of children and adolescents diagnosed with ASD, of which 32 were family informants (31.4%) and 70 professional informants in the educational field (68.6%). Among children and adolescents, they were 84.4% boys (*n* = 84) and 17.6% women (*n* = 18); 51 (50%) of them were in the stage of childhood and another 51 (50%) were in adolescence. Children and adolescents were aged between 6 and 18 and the mean age was 11.63 years (SD ± 3. 64). Table 1 shows sociodemographic information about the participants.

The descriptive statistics of the BRIEF-2 scale according to the family and school groups are shown in Table 2.

When comparing the mean scores obtained by children and adolescents with ASD according to informants, relatives, and education professionals, clinical differences are observed in the self-supervision, emotional control, initiative, and emotional regulation index of the BRIEF-2 scale.

The results of the two groups reflect a clinically significant range (T ≥ 70) in the self-supervision scale and in the emotional control index; whereas in the emotional control scale, the professionals yield potentially clinical elevation range (T 65–69) and the parents’ data reflect a slight elevation range (T 60–64). Finally, in the initiative scale, results are contrary to the previous ones; the data provided by the professionals show a mild clinical elevation (T 60–64), whereas the data of the parents reflect a potentially clinical elevation (T 65–69).

### 3.1. Differences between Groups: Family/School

The Mann–Whitney U shows significant differences in EFs between groups of family members and education professionals for the clinical scales of self-supervision, emotional control, and initiative and for the emotional regulation index of the BRIEF-2 scale. The rest showed no significant differences (Table 3).

### 3.2. Relationship between Regulatory Indices and Clinical Scales

The indices of behavioral regulation, emotional regulation, and cognitive regulation each correlate with the respective clinical scales that compose them for both groups (Table 4 and Table 5). All correlations found are positive.

Table 3 shows, in addition to the correlations of each index with the scales that form it, in the group of family informants, the indices of behavioral regulation and cognitive regulation correlate with the rest of the clinical scales, except flexibility. The emotional regulation index correlates with inhibition and self-supervision but not with the rest.

In the group of professional informants at school, the behavioral regulation index correlates with all clinical scales, in addition to those that make up each of them. For the emotional regulation index, correlation appears with the rest of the clinical scales, except working memory and task supervision. The cognitive regulation index correlates with all clinical scales except emotional control (Table 4).

### 3.3. Relationship between Regulatory Indices and Age

There is no significant correlation between age and the score of the different variables, (0.895 < *p* > 0.95), obtaining the approximate value to the correlation in supervision of the task, in the family, nor in the school (0.962 < *p* > 0.074), obtaining the highest correlation in planning and organization. Assuming a significance of 0.1, we can say that there is a positive correlation (r(32) = 0.300, *p* = 0.095) between age and task supervision, so, the older they are, the more they score in task supervision, within the family context. Furthermore, there is a negative correlation (r(70) = −0.215, *p* =0.074) between planning and organization and age at school environment: the older they are, the lower they score.

### 3.4. Differences by Sex

Regarding sex, there are no significant differences in the family, with 0.895 > *p* > 0.095 being the biggest difference in emotional control, where girls score more (68) than boys (60). In the school environment, there are significant differences according to sex, in the following values: self-supervision (*p* = 0.045), planning and organization (*p* = 0.009), behavioral regulation index (*p* = 0.023), emotional regulation index (*p* = 0.042), and executive regulation index (*p* = 0.011), with women scoring higher in all cases.

## 4. Discussion

The objective of this study was to analyze the level of development of EF in children and adolescents with autism in two social contexts, familial and educational, through the perception of informants related to the two environments (parents and professionals in the school environment). 

The results of the BRIEF-2 questionnaire show data with apparent clinical relevance (T > 59) on every scale and index, indicating that children and adolescents with ASD have problems in executive functioning [4,6,17,30]. When verifying the comparison between informants, the results reflected statistically significant differences in the scales, self-supervision, emotional control, and initiative, as well as in the emotional control index. Although the discrepancy in T scores between informants differs by less than 10 points on all scales and indices and could suggest the non-existence of differences of views in all cases [28], education professionals obtained higher scores than parents, except in initiative. These discrepancies in the data can be significant and complex to elucidate because it is not only the different daily situations that influence children’s and adolescents’ behavior but also abstract aspects, such as the expectations of the informants or the perceptual differences of the evaluators acquiring relevance [11].

According to the Bausela-Herreras study [6], with regard to the scales related to behavior and emotion (self-supervision and emotional control), education professionals obtained higher ranks, manifesting a greater degree of difficulty in the EF of those evaluated compared to the results of family members. This finding may suggest that professionals are more familiar with the functional arrangements for the age of the people evaluated, allowing them to recognize better the difficulties within these executive processing domains.

The self-supervision scale, which values one’s own behavior in relation to the rules, allows for knowledge of personal skills and abilities to be decisive in everyday situations and learn from experience. The score of professionals (T 76.04) is higher than the score of parents (T 71.03), but in both cases it is clinically significant (T ≥ 70); this high score in both contexts suggests the low awareness of children and adolescents with ASD before the impact and consequences that their behavior can exert on others [28]. The literature provides studies on the difficulties of people with autism when inhibiting automatic responses [9,14,15,16,17,18,19], aspects closely related to self-supervision. 

With regard to the emotional control scale, it acquires great importance in the correct functioning of the EF [31], because it is related to difficulties when modulating responses adapted to different contexts. Following the line of the study by Blijd et al. [4], family informants were evaluated below the clinical limit (T 62); in contrast, in the school context, they achieved potentially clinical scores (T 69.91). These findings suggest that education professionals are either more sensitive or give more importance to the possible emotional lability in people with ASD in a context where it may be more complicated to control their emotional states and, at the same time, respond to exaggerated behaviors and to the demands of the environment, such as emotional explosiveness.

On the other hand, the emotional control index brings together the scales of flexibility and emotional control and is related to the degree of difficulty in regulating adapted emotional responses when situations change in different contexts; the data of the study with clinical significance are very high both in the family (T 70.68) and in school (T 77.62), so they suggest that the cognitive flexibility component is a fundamental gradient in the difficulties in EF, contemplated in various studies [4,5,6,12,13,14,15,16,17,18,19].

Emotional control may be a precedent for adequate cognitive regulation [32]. In the present study, only the scale of initiative is statistically significant. Unlike those related to behavior and emotion, on this metacognition scale, the perception of informants is inverse, where there is a tendency of parents to observe greater difficulties [11,33]. The data of the family group reflects a potentially clinical elevation (T 69.18) against the perception of the school that remains in mild elevation (T 64.24). These results suggest, on the one hand, that a more structured and less flexible and tolerant school environment, with proposed and controlled activities, can influence the initiative of children and adolescents more than in the home context, where parents can give more importance to the performance of daily activities more independently with additional factors (such as that they arise from the child and are not so directed). 

Taking into account that this scale reflects the possible problem of initiating activities implicitly entailing an ideation, planning, and execution for any act of occupation autonomously, this dimension of EF can be related primarily, with difficulties in other dimensions of the EF, such as, for example, in the organization or planning of the activities [4,27] or secondarily with emotional problems, depressive states, or oppositional behaviors [28].

The comparison of results between informants/parents and informants/professionals suggests the need to develop specific skills related to EF to adapt to different environments. Children and adolescents in the school context are presented with new concepts more frequently than in the family context, which suggests that EF related to response inhibition or cognitive flexibility are developed more in this environment [34], in order to be able to adapt their behavioral responses effectively to a more changing environment.

Regarding the concordance between scales and indices according to informants, the findings are disparate, depending on whether the informant is a parent or an education professional. Even so, difficulties are contemplated at the level of executive functioning in general.

In the case of parents, the findings with the emotional regulation index are striking, because there is no relationship with any of the metacognition scales, or, specifically, the flexibility scale with the behavioral regulation index and the cognitive regulation index; and, on the other hand, how the emotional control scale correlates to a greater degree with the behavioral regulation index than with the cognitive regulation index. These preliminary discoveries could be related to the exceptional circumstances that are being experienced, due to the COVID-19 pandemic, in which parents contemplate their children with a greater degree of “rigidity”, with the need to maintain more established routines or with an inability to change activities. They cling to their normally very restricted interests and, therefore, give as answers behaviors not adapted to the environment, so that the informants of the family environment give greater emphasis to the emotional aspects than to the cognitive ones. These data support the idea of how behavior may affect more cognitive functions [28].

In the case of education professionals, they practically correlate all scales with each other, and although the findings indicate a low relationship strength on some scales, it suggests that professionals are more sensitive to the development of EF than parents [6,11]. In the same line as the results of the parents, when analyzing the relationship between the behavioral regulation index with the emotional scales and the emotional regulation index with the behavioral scales, there is a high correlation between the dimensions, suggesting that regardless of the informant, good regulation of both behavior and emotion is essential for proper executive functioning.

The use of data from different sources and contexts increases the accuracy of the information received by admitting the triangulation of sources [33]. Although EF deficits are common in children and adolescents with ASD, they are not part of the diagnostic criteria, but are related to adaptive behavior [35], and can be important predictors of adaptive behavior [36] and influence the quality of life of people with ASD [37]. 

Discrepancies between informants should not be ignored as they may reflect real differences in behavior [38]. Behavioral assessment instruments of EF, with ecological validity, can provide significant information on how difficulties in EF affect daily performance [21,39,40]. Clinically, it is important to understand the different behavioral responses in childhood and adolescence in different contexts and that these can affect the intervention, which is why it is so important to implement interventions in natural contexts [41]. Based on executive functioning, this benefits people with early difficulties, by solving the low academic performance during the school stage [3].

The lack of statistically significant correlations between age and questionnaire scales can be attributed to the limited sample size. Although the results are not consistent, it should be noted that the correlation between ages only shows an approach in the responses of family members, increasing the perception of difficulties in supervising the task under development [25]. Whereas the results of professionals on the scale of planning and organization are inverse; this way it seems that difficulties in planning and organization decrease [18]. This may be due to a perception of developmental maturity on the part of professionals and not by family members on the one hand or to the use of compensatory strategies [21] in the school environment, even if executive functioning problems persist.

It is significant to check comparisons by sex, where women with ASD performed on average significantly worse in EF than men, being that these differences are attributable to all regulatory indices in the school context. These findings may reflect the potential difficulties in all dimensions (behavioral, emotional, and cognitive) of women in school, whereas in a friendlier and less social context such as the home, the difficulties in EF are lower and they are in the same line as men. A study developed by White et al. [42] found that parents rated females with greater problems with executive function also using the BRIEF tool; their results, as in the present study, show relative EF weaknesses for females compared to males, so this may have important implications for planning and implementing treatments. Research on EF in women with ASD is scarce, which, although the results cannot be generalized, is a novel aspect of this study.

The fundamental contribution of this research lies in the fact that it develops the executive function in two fundamental contexts of the child with ASD, such as school and family, and takes into account various variables, such as sex and age, obtaining fundamental data for the intervention and development of different strategies according to sex, age, and context.

It is important to note the limitations of this study. First of all, the type of study that has been performed does not allow one to make causal relationships. The selected sample is small, and, in addition, there is a difference between the selected groups, because in one there are 32 informants who are parents and in the other, there are 70 educational professionals. This also reflects on effect size and statistical power that offer small to medium results in most cases; so, these preliminary results cannot be generalized. 

With regard to future lines of research, in the present study a series of demographic factors related to children and adolescents with ASD were controlled, but specific factors about the informants were not investigated, such as their gender, the amount of time that the informant-professional knew the person with ASD, or the possible level of stress. These are variables that can affect the results [43] and that need further research. 

Another future line of research could be the knowledge of the degree of agreement between informants, which in this study has not been able to be carried out with the data obtained because they are not coincidental, and which has been investigated in multiple populations [44,45,46,47,48,49].

The limitations could be addressed in the future with a larger and more balanced sample size and perhaps a wider age range, which would give access to comparisons by age groups to determine if the reliability between informants changes through the life cycle. A calculation of the necessary sample size made using the G-Power statistical program establishes that, for an average effect size (Effect size d 0.5), the sample would be composed of 67 people in each group, that is, 134 informants in total, which would also compensate for the groups’ imbalance. 

It is also important to compile information from several informants to reach a complete profile that helps in the design and implementation of interventions that benefit people with early difficulties and solves the low academic performance that may occur during the school stage.

## 5. Conclusions

This research highlights the differences found in the variables of self-monitoring, emotional control, initiative, and emotional regulation index between school and family environments, with worse executive function in school. This must be considered in order to generate appropriate intervention techniques according to the context. A good executive functioning is associated with correct emotional and behavioral regulation, which optimizes daily performance.

Regarding sex, significant differences have been found in various variables, in self-supervision, planning and organization, behavioral regulation index, emotional regulation index, and in the executive regulation index, with women scoring higher in all of them.

There are no significant correlations with age, although it can be pointed out that there is a positive correlation between age and supervision of the task, that is, the older individuals become, the greater they score in supervision of the task, within the family. Furthermore, there is a negative correlation within the school between planning and organization and age, that is, the older individuals become, the lower they score.

## Figures and Tables

**Table 1 ijerph-19-07834-t001:** Sociodemographic data.

		Frequency	Percentage	Valid Percentage	Accumulated Percentage
Group	Family	32	31.4	31.4	31.4
	School	70	68.6	68.6	100.0
Sex (child or adolescent)	Male	84	82.4	82.4	82.4
	Female	18	17.6	17.6	100.0
Age (range)	Childhood	51	50.0	50.0	50.0
	Adolescence	51	50.0	50.0	100.0
School stage	Primary education	53	52.0	52.0	52.0
	Secondary education	49	48.0	48.0	100.0

**Table 2 ijerph-19-07834-t002:** Executive functions’ descriptive data for family/school.

	Group	*n*	Mean	SD	SEM
T Inhibition	Family	32	65.62	15.06	2.66
	School	70	68.21	13.57	1.62
T Self-monitoring	Family	32	71.03	10.27	1.81
	School	70	76.04	12.73	1.52
T Flexibility	Family	32	76.00	15.24	2.69
	School	70	79.18	12.05	1.44
T Emotional control	Family	32	62.00	11.66	2.06
	School	70	69.91	15.72	1.87
T Initiative	Family	32	69.18	10.16	1.79
	School	70	64.24	9.91	1.18
T Working memory	Family	32	66.31	11.52	2.03
	School	70	66.58	10.01	1.19
T Planning and organization	Family	32	65.50	11.47	2.02
	School	70	68.87	10.18	1.21
T Task monitoring	Family	32	64.06	10.78	1.90
	School	70	65.45	10.41	1.24
T Organization of materials	Family	32	64.40	17.68	3.12
	School	70	63.01	13.17	1.57
T Behavioral Regulation Index	Family	32	69.43	13.06	2.30
	School	70	72.50	12.26	1.46
T Emotional Regulation index	Family	32	70.68	13.72	2.42
	School	70	77.62	14.26	1.70
T Cognitive Regulation index	Family	32	69.09	11.42	2.02
	School	70	67.91	10.00	1.19
T Executive Regulation Index	Family	32	72.78	12.23	2.16
	School	70	74.08	10.58	1.26

T: in T scores of the general population differentiated by sex, age range, and informants; SD: Standard deviation; SEM: Standard error mean.

**Table 3 ijerph-19-07834-t003:** Executive function comparison between groups. Mann–Whitney U.

Variable	Median Family	Median School	Mann–Whitney U	Z	*p*-Value	1-β	*d*
T Inhibition	65.00	67.50	1006.00	−0.823	0.411	0.55	0.18
T Self-monitoring	73.00	79.00	827.00	−2.118	**0.034**	0.43	0.43
T Flexibility	78.00	81.50	981.00	−1.003	0.316	0.54	0.23
T Emotional control	65.00	71.50	736.00	−2.771	**0.006**	0.42	0.57
T Initiative	69.50	66.00	822.00	−2.152	**0.031**	0.52	0.49
T Working memory	67.00	65.00	1108.50	−2.152	0.934	0.93	0.02
T Planning and organization	68.50	68.00	954.00	−1.199	0.231	0.59	0.31
T Task monitoring	64.50	68.00	1071.50	−0.350	0.726	0.76	0.13
T Organization of materials	65.50	64.50	1050.50	−0.502	0.616	0.64	0.08
T Behavioral Regulation Index	69.00	71.50	956.50	−1.180	0.238	0.48	0.24
T Emotional Regulation index	68.50	79.50	806.00	−2.266	**0.023**	0.48	0.49
T Cognitive Regulation index	69.50	68.50	1034.00	−0.621	0.535	0.58	0.10
T Index of ejective regulation	71.00	75.00	1044.50	−0.545	0.586	0.63	0.11

T: T scores of the general population differentiated by sex, age range, and informants. Bold: significant values.

**Table 4 ijerph-19-07834-t004:** Correlations of family. Spearman’s Rho.

		T Behavioral Regulation Index	T Emotional Regulation Index	T Cognitive Regulation Index
T Inhibition	Corr. Coeff.	0.944 **	0.584 **	0.710 **
	Sig. (2-tailed)	**<0.001**	**<0.001**	**<0.001**
T Self-monitoring	Corr. Coeff.	0.731 **	0.361 *	0.590 **
	Sig. (2-tailed)	**<0.001**	**0.042**	**<0.001**
T Flexibility	Corr. Coeff.	0.43	0.872 **	0.308
	Sig. (2-tailed)	0.055	**<0.001**	0.086
T Emotional control	Corr. Coeff.	0.715 **	0.896 **	0.445 *
	Sig. (2-tailed)	**<0.001**	**<0.001**	**0.011**
T Initiative	Corr. Coeff.	0.463 **	0.252	0.675 **
	Sig. (2-tailed)	**0.008**	0.163	**<0.001**
T Working memory	Corr. Coeff.	0.605 **	0.285	0.892 **
	Sig. (2-tailed)	**<0.001**	0.114	**<0.001**
T Planning and organization	Corr. Coeff.	0.773 **	0.302	0.927 **
	Sig. (2-tailed)	**<0.001**	0.093	**<0.001**
T Task monitoring	Corr. Coeff.	0.497 **	0.242	0.634 **
	Sig. (2-tailed)	**0.004**	0.182	**<0.001**
T Organization of materials	Corr. Coeff.	0.574 **	0.234	0.762 **
	Sig. (2-tailed)	**<0.001**	0.198	**<0.001**

T: in T scores of the general population differentiated by sex, age range, and informants. Bold: significant values. * Correlation is significant at the 0.05 level (2-sided). ** Correlation is significant at the 0.01 level (2-sided).

**Table 5 ijerph-19-07834-t005:** Correlations at school. Spearman’s Rho.

		T T Behavioral Regulation Index	T Emotional Regulation Index	T Cognitive Regulation Index
T Inhibition	Corr. Coeff.	0.920 **	0.572 **	0.297 *
	Sig. (2-tailed)	**<0.001**	**<0.001**	**0.012**
T Self-monitoring	Corr. Coeff.	0.720 **	0.624 **	0.479 **
	Sig. (2-tailed)	**<0.001**	**<0.001**	**<0.001**
T Flexibility	Corr. Coeff.	0.541 **	0.851 **	0.372 **
	Sig. (2-tailed)	**<0.001**	**<0.001**	**0.001**
T Emotional control	Corr. Coeff.	0.713 **	0.916 **	0.206
	Sig. (2-tailed)	**<0.001**	**<0.001**	0.088
T Initiative	Corr. Coeff.	0.241 *	0.282 *	0.764 **
	Sig. (2-tailed)	**0.044**	**0.018**	**<0.001**
T Working memory	Corr. Coeff.	0.316 **	0.195	0.886 **
	Sig. (2-tailed)	**0.008**	0.106	**<0.001**
T Planning and organization	Corr. Coeff.	0.423 **	0.323 **	0.874 **
	Sig. (2-tailed)	**<0.001**	**0.006**	**<0.001**
T Task monitoring	Corr. Coeff.	0.311 **	0.175	0.878 **
	Sig. (2-tailed)	**0.009**	0.147	**<0.001**
T Organization of materials	Corr. Coeff.	0.390 **	0.271 *	0.716 **
	Sig. (2-tailed)	**<0.001**	**0.023**	**<0.001**

T: in T scores of the general population differentiated by sex, age range, and informants. Bold: significant values. * Correlation is significant at the 0.05 level (2-sided). ** Correlation is significant at the 0.01 level (2-sided).

## Data Availability

Not applicable.

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
