# Peer review of "Executive Functions in Children and Adolescents with Autism Spectrum Disorder in Family and School Environment"

_ijerph, 2022, doi:10.3390/ijerph19137834_

Round 1

Reviewer 1 Report

In the current study authors aimed to analyze the level of development of the Executive Function (EF) in children and adolescents with autism between 6-18 years old from educational and family context.

The study has the following concerns:

1. Authors do not provide enough proofs for understanding the role of EF in ASD

2. There is a big gap between the age groups of ASD children and it is not clear how many participants (concerning children and adolescents) were in concrete group.

3. Statistics is not well balanced - 32 patients of children with ASD and 70 professionals.

4. There was only one method - the questionnaire (BRIEF-2) used in research, which is not enough for making strong judgements.

5. In "Discussion" as well as "Conclusion" part outcomes are very descriptive, thoughts and conclusions do not follow from the data obtained.

I would advice authors to make additional tests, checks and analyzes for the completeness of the study.

Reviewer 2 Report

Thank you for the opportunity to read this study. I think this manuscript is relevant, but I find several problems:

  • There are some mistakes concerning compliance with the journal's guidelines: the abstract maintains the labels, tables should also be revised, and the list of references is not the one the journal recommends…
  • One of my biggest concerns with this work lies in its novelty. We already know that executive dysfunction includes in the occupations and in the daily life of people with autism. So, I would like to ask the authors what contributions to science this article brings to the field?
    • https://www.nature.com/articles/mp201775
    • https://www.frontiersin.org/articles/10.3389/fpsyt.2019.00753/full
    • https://www.sciencedirect.com/science/article/pii/S1750946721000829
    • https://www.dovepress.com/a-review-of-executive-function-deficits-in-autism-spectrum-disorder-an-peer-reviewed-fulltext-article-NDT
    • https://pubmed.ncbi.nlm.nih.gov/30418493/
    • https://www.researchgate.net/publication/246154674_The_contribution_of_executive_functions_to_participation_in_school_activities_of_children_with_high_functioning_autism_spectrum_disorder
    • https://www.researchgate.net/publication/316461999_Autism_spectrum_disorders_A_meta-analysis_of_executive_function
    • https://pubmed.ncbi.nlm.nih.gov/30847709/
    • https://journals.sagepub.com/doi/10.1177/2396941518800775
    • https://www.neurologia.com/articulo/2019133/eng
    • https://link.springer.com/article/10.1007/s10803-014-2071-4
  • Writing and translation into English contain some mistakes. It should be reviewed.
  • The text has severe problems with internal coherence. As you read, "it seems to go one way" and then "it goes the another".
  • Introduction: the writing is extremely poor and poorly organised. It is composed of short, unconnected paragraphs. Please explain all the characteristics of the study variables and how they influence the daily life of children and adolescents...
  • Part of the design type subsection information is part of the procedure.
  • Who do the authors consider to be the participants? If they are children, they should always be signed by parents or legal guardians. If only 32 were signed, how could there be 102 participants? It would be unethical. On the other hand, if the participants are parents and teachers, this should be explained and include the eligibility criteria for each case. In addition, information on parent and teacher participation should be introduced in the introduction in a much more precise way.
  • A procedure subsection is missing. Detailed information on the whole procedure should be given so that other researchers can replicate their study.
  • Please include the calculation of the sample size and power of the study.
  • Was your study registered?
  • The statistics section is poor. The statistics and criteria used to determine and interpret their results are indicated.
  • The EFs develop as they grow older, into adulthood. Comparisons between critical ages would be advisable. This information neither is in the introduction.
  • As the authors describe that the data do not comply with normality, and I am not sure about that. I am asking for more information about the adjustment or non-adjustment to normality or providing me with the data to carry out the corresponding tests.
  • Looking at the results and considering that the aim is not to measure EF in children and adolescents but to check whether there are differences between parents or teachers, the groups were not balanced. This, too, detracts from the credibility of the results.
  • A table with concrete socio-demographic information in the participants' section would be helpful.
  • All abbreviations should be explained. No abbreviations may appear in table titles. All abbreviations in the table must appear in the table footer.
  • In addition to comparing your results with other studies and looking for relationships, the discussion should include implications, practices, limitations, and future directions. Its limitations are minimal, and its study presents many more problems than described. With the type of study that has been conducted, they cannot make causal relationships.
  • "In conclusion, although EF deficits are not integrated into the diagnostic criteria". This is not entirely true. They may not be mentioned directly, but they are included in restrictive behaviours. Conclusions should only refer to the results of your study.

The main problems are my the doubts regarding its originality and procedures and statistical test applied and the remarkable lack of internal coherence in the text.

Reviewer 3 Report

This is a descriptive, cross-sectional, multicentre study examining 102 participants selected by non-probabilistic sampling, 32 parents of children with ASD  and 70 professionals in the field of education of students with ASD. The results of the study confirm that children and adolescents with ASD present problems in executive functioning, but the perception of informants, parents and education professionals, is similar but not the same in the different contexts, school and home.

The data are clearly reported and discussed. English style is adequate, such as statistical analysis.

There is a large amount of literature evidence on executive functions in ASD. I think that the Authors should specify and underline in the Discussion what is the innovative information of this paper in comparison with similar studies in literature.

Round 2

Reviewer 1 Report

Dear Authors,

After careful revision of the Manuscript once again, look through the changes you have done, in my opinion, the current research needs more data, which could give valuable proofs for completeness of the study and strong judgements.   

Author Response

Dear Reviewer,

We appreciate your comments, which are undoubtedly of great importance to improve the quality of the article.

In this second review, age and gender data have been added, which have  completed the study.

We will take it into account for future studies, and we add it in limitations. Thank you very much

Reviewer 2 Report

I would like to thank the authors for this new version and their efforts to improve their manuscript, whose quality has undoubtedly improved. Please review these issues:

·      Significant issues persist with the English language.

·      References are still not presented as recommended by the journal.

·      Dots should replace all commas in tables.

·      All abbreviations should be explained the first time they appear both in the abstract and in the text, then you can use them freely. All abbreviations in the table must appear in the table footer. 

·      The BRIEF explanation in the introduction is a bit forced, as there are several methodologies and multiple tools to measure executive function. Don't the authors consider that they duplicate information contained in the methods section?

·      The research question should close the introduction, along with the aim.

·      What is the meaning of the T in Table 2 and further?

·      You don't need to repeat "Corr. Coeff. Sig. (2-tailed) <,001" in the tables. You can add a footnote.

·      Conclusions should be brief, answering the research question/goal.

·      Perhaps it would be cautious about using the term pilot study or preliminary results due to the study's limitations.
